# Jacobian Adversarially Regularized Networks for Robustness

**Alvin Chan**[1][*], **Yi Tay**[1], **Yew-Soon Ong**[1], **Jie Fu**[2]
[1]Nanyang Technological University,   [2]Mila, Polytechnique Montreal

## Abstract

Adversarial examples are crafted with imperceptible perturbations with the intent to fool neural networks. Against such attacks, adversarial training and its variants stand as the strongest defense to date. Previous studies have pointed out that robust models that have undergone adversarial training tend to produce more salient and interpretable Jacobian matrices than their non-robust counterparts. A natural question is whether a model trained with an objective to produce salient Jacobian can result in better robustness. This paper answers this question with affirmative empirical results. We propose Jacobian Adversarially Regularized Networks (JARN) as a method to optimize the saliency of a classifier's Jacobian by adversarially regularizing the model's Jacobian to resemble natural training images[1]. Image classifiers trained with JARN show improved robust accuracy compared to standard models on the MNIST, SVHN and CIFAR-10 datasets, uncovering a new angle to boost robustness without using adversarial training examples.

## 1 Introduction

Deep learning models have shown impressive performance in a myriad of classification tasks (LeCun et al., 2015). Despite their success, deep neural image classifiers are found to be easily fooled by visually imperceptible adversarial perturbations (Szegedy et al., 2013). These perturbations can be crafted to reduce accuracy during test time or veer predictions towards a target class. This vulnerability not only poses a security risk in using neural networks in critical applications like autonomous driving (Bojarski et al., 2016) but also presents an interesting research problem about how these models work.

Many adversarial attacks have come into the scene (Carlini & Wagner, 2017; Papernot et al., 2018; Croce & Hein, 2019), not without defenses proposed to counter them (Gowal et al., 2018; Zhang et al., 2019). Among them, the best defenses are based on *adversarial training* (AT) where models are trained on adversarial examples to better classify adversarial examples during test time (Madry et al., 2017). While several effective defenses that employ adversarial examples have emerged (Qin et al., 2019; Shafahi et al., 2019), generating strong adversarial training examples adds non-trivial computational burden on the training process (Kannan et al., 2018; Xie et al., 2019).

Adversarially trained models gain robustness and are also observed to produce more salient Jacobian matrices (Jacobians) at the input layer as a side effect (Tsipras et al., 2018). These Jacobians visually resemble their corresponding images for robust models but look much noisier for standard non-robust models. It is shown in theory that the saliency in Jacobian is a result of robustness (Etmann et al., 2019). A natural question to ask is this: can an improvement in Jacobian saliency induce robustness in models? In other words, could this side effect be a new avenue to boost model robustness? To the best of our knowledge, this paper is the first to show affirmative findings for this question.

To enhance the saliency of Jacobians, we draw inspirations from neural generative networks (Choi et al., 2018; Dai & Wipf, 2019). More specifically, in generative adversarial networks (GANs) (Goodfellow et al., 2014), a generator network learns to generate natural-looking images with a training objective to fool a discriminator network. In our proposed approach, Jacobian Adversarially

---

[*]Corresponding author: `guoweial001@ntu.edu.sg`
[1]Source code available at `https://github.com/alvinchangw/JARN_ICLR2020`

Regularized Networks (JARN), the classifier learns to produce salient Jacobians with a regularization objective to fool a discriminator network into classifying them as input images. This method offers a new way to look at improving robustness without relying on adversarial examples during training. With JARN, we show that directly training for salient Jacobians can advance model robustness against adversarial examples in the MNIST, SVHN and CIFAR-10 image dataset. When augmented with adversarial training, JARN can provide additive robustness to models thus attaining competitive results. All in all, the prime contributions of this paper are as follows:

- We show that directly improving the saliency of classifiers' input Jacobian matrices can increase its adversarial robustness.
- To achieve this, we propose Jacobian adversarially regularized networks (JARN) as a method to train classifiers to produce salient Jacobians that resemble input images.
- Through experiments in MNIST, SVHN and CIFAR-10, we find that JARN boosts adversarial robustness in image classifiers and provides additive robustness to adversarial training.

## 2 BACKGROUND AND RELATED WORK

Given an input $\mathbf{x}$, a classifier $f(\mathbf{x}; \theta) : \mathbf{x} \mapsto \mathbb{R}^k$ maps it to output probabilities for $k$ classes in set $C$, where $\theta$ is the classifier's parameters and $\mathbf{y} \in \mathbb{R}^k$ is the one-hot label for the input. With a training dataset $D$, the standard method to train a classifier $f$ is empirical risk minimization (ERM), through $\min_\theta \mathbb{E}_{(\mathbf{x},\mathbf{y}) \sim D} \mathcal{L}(\mathbf{x}, \mathbf{y})$, where $\mathcal{L}(\mathbf{x}, \mathbf{y})$ is the standard cross-entropy loss function defined as

$$\mathcal{L}(\mathbf{x}, \mathbf{y}) = \mathbb{E}_{(\mathbf{x},\mathbf{y}) \sim D} \left[ -\mathbf{y}^\top \log f(\mathbf{x}) \right] \tag{1}$$

While ERM trains neural networks that perform well on holdout test data, their accuracy drops drastically in the face of adversarial test examples. With an adversarial perturbation of magnitude $\varepsilon$ at input $\mathbf{x}$, a model is robust against this attack if

$$\arg\max_{i \in C} f_i(\mathbf{x}; \theta) = \arg\max_{i \in C} f_i(\mathbf{x} + \delta; \theta) , \quad \forall \delta \in B_p(\varepsilon) = \delta : \|\delta\|_p \leq \varepsilon \tag{2}$$

We focus on $p = \infty$ in this paper.

**Adversarial Training** To improve models' robustness, adversarial training (AT) (Goodfellow et al., 2016) seek to match the training data distribution with the adversarial test distribution by training classifiers on adversarial examples. Specifically, AT minimizes the loss function:

$$\mathcal{L}(\mathbf{x}, \mathbf{y}) = \mathbb{E}_{(\mathbf{x},\mathbf{y}) \sim D} \left[ \max_{\delta \in B(\varepsilon)} \mathcal{L}(\mathbf{x} + \delta, \mathbf{y}) \right] \tag{3}$$

where the inner maximization, $\max_{\delta \in B(\varepsilon)} \mathcal{L}(\mathbf{x} + \delta, \mathbf{y})$, is usually performed with an iterative gradient-based optimization. Projected gradient descent (PGD) is one such strong defense which performs the following gradient step iteratively:

$$\delta \leftarrow \mathrm{Proj} \left[ \delta - \eta \, \mathrm{sign} \left( \nabla_\delta \mathcal{L}(\mathbf{x} + \delta, \mathbf{y}) \right) \right] \tag{4}$$

where $\mathrm{Proj}(\mathbf{x}) = \arg\min_{\zeta \in B(\varepsilon)} \| \mathbf{x} - \zeta \|$. The computational cost of solving Equation (3) is dominated by the inner maximization problem of generating adversarial training examples. A naive way to mitigate the computational cost involved is to reduce the number gradient descent iterations but that would result in weaker adversarial training examples. A consequence of this is that the models are unable to resist stronger adversarial examples that are generated with more gradient steps, due to a phenomenon called obfuscated gradients (Carlini & Wagner, 2017; Uesato et al., 2018).

Since the introduction of AT, a line of work has emerged that also boosts robustness with adversarial training examples. Capturing the trade-off between natural and adversarial errors, TRADES (Zhang et al., 2019) encourages the decision boundary to be smooth by adding a regularization term to reduce the difference between the prediction of natural and adversarial examples. Qin et al. (2019)

seeks to smooth the loss landscape through local linearization by minimizing the difference between the real and linearly estimated loss value of adversarial examples. To improve adversarial training, Zhang & Wang (2019) generates adversarial examples by feature scattering, i.e., maximizing feature matching distance between the examples and clean samples.

Tsipras et al. (2018) observes that adversarially trained models display an interesting phenomenon: they produce salient Jacobian matrices ($\nabla_{\mathbf{x}}\mathcal{L}$) that loosely resemble input images while less robust standard models have noisier Jacobian. Etmann et al. (2019) explains that linearized robustness (distance from samples to decision boundary) increases as the alignment between the Jacobian and input image grows. They show that this connection is strictly true for linear models but weakens for non-linear neural networks. While these two papers show that robustly trained models result in salient Jacobian matrices, our paper aims to investigate whether directly training to generate salient Jacobian matrices can result in robust models.

**Non-Adversarial Training Regularization**    Provable defenses are first proposed to bound minimum adversarial perturbation for certain types of neural networks (Hein & Andriushchenko, 2017; Weng et al., 2018; Raghunathan et al., 2018). One of the most advanced defense from this class of work (Wong et al., 2018) uses a dual network to bound the adversarial perturbation with linear programming. The authors then optimize this bound during training to boost adversarial robustness. Apart from this category, closer to our work, several works have studied a regularization term on top of the standard training objective to reduce the Jacobian's Frobenius norm. This term aims to reduce the effect input perturbations have on model predictions. Drucker & Le Cun (1991) first proposed this to improve model generalization on natural test samples and called it 'double backpropagation'. Two subsequent studies found this to also increases robustness against adversarial examples Ross & Doshi-Velez (2018); Jakubovitz & Giryes (2018). Recently, Hoffman et al. (2019) proposed an efficient method to approximate the input-class probability output Jacobians of a classifier to minimize the norms of these Jacobians with a much lower computational cost. Simon-Gabriel et al. (2019) proved that double backpropagation is equivalent to adversarial training with $l_2$ examples. Etmann et al. (2019) trained robust models using double backpropagation to study the link between robustness and alignment in non-linear models but did not propose a new defense in their paper. While the double backpropagation term improves robustness by reducing the effect that perturbations in individual pixel have on the classifiers prediction through the Jacobians norm, it does not have the aim to optimize Jacobians to explicitly resemble their corresponding images semantically. Different from these prior work, we train the classifier with an adversarial loss term with the aim to make the Jacobian resemble input images more closely and show in our experiments that this approach confers more robustness.

## 3    JACOBIAN ADVERSARIALLY REGULARIZED NETWORKS (JARN)

**Motivation**    Robustly trained models are observed to produce salient Jacobian matrices that resemble the input images. This begs a question in the reverse direction: will an objective function that encourages Jacobian to more closely resemble input images, will standard networks become robust? To study this, we look at neural generative networks where models are trained to produce natural-looking images. We draw inspiration from generative adversarial networks (GANs) where a generator network is trained to progressively generate more natural images that fool a discriminator model, in a min-max optimization scenario (Goodfellow et al., 2014). More specifically, we frame a classifier as the generator model in the GAN framework so that its Jacobians can progressively fool a discriminator model to interpret them as input images.

Another motivation lies in the high computational cost of the strongest defense to date, adversarial training. The cost on top of standard training is proportional to the number of steps its adversarial examples take to be crafted, requiring an additional backpropagation for each iteration. Especially with larger datasets, there is a need for less resource-intensive defense. In our proposed method (JARN), there is only one additional backpropagation through the classifier and the discriminator model on top of standard training. We share JARN in the following paragraphs and offer some theoretical analysis in § 3.1.

**Jacobian Adversarially Regularized Networks** Denoting input as $\mathbf{x} \in \mathbb{R}^{hwc}$ for $h \times w$-size images with $c$ channels, one-hot label vector of $k$ classes as $\mathbf{y} \in \mathbb{R}^k$, we express $f_{\text{cls}}(\mathbf{x}) \in \mathbb{R}^k$ as the prediction of the classifier ($f_{\text{cls}}$), parameterized by $\theta$. The standard cross-entropy loss is

$$\mathcal{L}_{\text{cls}} = \mathbb{E}_{(\mathbf{x},\mathbf{y})} \left[ -\mathbf{y}^\top \log f_{\text{cls}}(\mathbf{x}) \right] \tag{5}$$

With gradient backpropagation to the input layer, through $f_{\text{cls}}$ with respect to $\mathcal{L}_{\text{cls}}$, we can get the Jacobian matrix $J \in \mathbb{R}^{hwc}$ as:

$$J(\mathbf{x}) := \nabla_{\mathbf{x}} \mathcal{L}_{\text{cls}} = \left[ \frac{\partial \mathcal{L}_{\text{cls}}}{\partial \mathbf{x}_1} \quad \cdots \quad \frac{\partial \mathcal{L}_{\text{cls}}}{\partial \mathbf{x}_d} \right] \tag{6}$$

where $d = hwc$. The next part of JARN entails adversarial regularization of Jacobian matrices to induce resemblance with input images. Though the Jacobians of robust models are empirically observed to be similar to images, their distributions of pixel values do not visually match (Etmann et al., 2019). The discriminator model may easily distinguish between the Jacobian and natural images through this difference, resulting in the vanishing gradient (Arjovsky et al., 2017) for the classifier train on. To address this, an adaptor network ($f_{apt}$) is introduced to map the Jacobian into the domain of input images. In our experiments, we use a single 1x1 convolutional layer with $tanh$ activation function to model $f_{apt}$, expressing its model parameters as $\psi$. With the $J$ as the input of $f_{\text{apt}}$, we get the adapted Jacobian matrix $J' \in \mathbb{R}^{hwc}$,

$$J' = f_{\text{apt}}(J) \tag{7}$$

We can frame the classifier and adaptor networks as a generator $G(\mathbf{x}, \mathbf{y})$

$$G_{\theta,\psi}(\mathbf{x}, \mathbf{y}) = f_{\text{apt}}( \nabla_{\mathbf{x}} \mathcal{L}_{\text{cls}}(\mathbf{x}, \mathbf{y}) ) \tag{8}$$

learning to model distribution of $p_{J'}$ that resembles $p_{\mathbf{x}}$.

We now denote a discriminator network, parameterized by $\phi$, as $f_{\text{disc}}$ that outputs a single scalar. $f_{\text{disc}}(\mathbf{x})$ represents the probability that $\mathbf{x}$ came from training images $p_{\mathbf{x}}$ rather than $p_{J'}$. To train $G_{\theta,\psi}$ to produce $J'$ that $f_{\text{disc}}$ perceive as natural images, we employ the following adversarial loss:

$$\begin{aligned} \mathcal{L}_{\text{adv}} &= \mathbb{E}_{\mathbf{x}}[\log f_{\text{disc}}(\mathbf{x})] + \mathbb{E}_{J'}[\log(1 - f_{\text{disc}}(J'))] \\ &= \mathbb{E}_{\mathbf{x}}[\log f_{\text{disc}}(\mathbf{x})] + \mathbb{E}_{(\mathbf{x},\mathbf{y})}[\log(1 - f_{\text{disc}}(G_{\theta,\psi}(\mathbf{x})))] \\ &= \mathbb{E}_{\mathbf{x}}[\log f_{\text{disc}}(\mathbf{x})] + \mathbb{E}_{(\mathbf{x},\mathbf{y})} [\log(1 - f_{\text{disc}}( f_{\text{apt}}( \nabla_{\mathbf{x}} \mathcal{L}_{\text{cls}}(\mathbf{x}, \mathbf{y}))))] \end{aligned} \tag{9}$$

Combining this regularization loss with the classification loss function $\mathcal{L}_{\text{cls}}$ in Equation (5), we can optimize through stochastic gradient descent to approximate the optimal parameters for the classifier $f_{\text{cls}}$ as follows,

$$\theta^* = \arg\min_{\theta}(\mathcal{L}_{cls} + \lambda_{adv}\mathcal{L}_{adv}) \tag{10}$$

where $\lambda_{adv}$ control how much Jacobian adversarial regularization term dominates the training.

Since the adaptor network ($f_{\text{apt}}$) is part of the generator $G$, its optimal parameters $\psi^*$ can be found with minimization of the adversarial loss,

$$\psi^* = \arg\min_{\psi} \mathcal{L}_{adv} \tag{11}$$

On the other hand, the discriminator ($f_{\text{disc}}$) is optimized to maximize the adversarial loss term to distinguish Jacobian from input images correctly,

$$\phi^* = \arg\max_{\phi} \mathcal{L}_{adv} \tag{12}$$

Analogous to how generator from GANs learn to generate images from noise, we add $[-\varepsilon, -\varepsilon]$ uniformly distributed noise to input image pixels during JARN training phase. Figure 1 shows a

summary of JARN training phase while Algorithm 1 details the corresponding pseudo-codes. In our experiments, we find that using JARN framework only on the last few epoch (25%) to train the classifier confers similar adversarial robustness compared to training with JARN for the whole duration. This practice saves compute time and is used for the results reported in this paper.

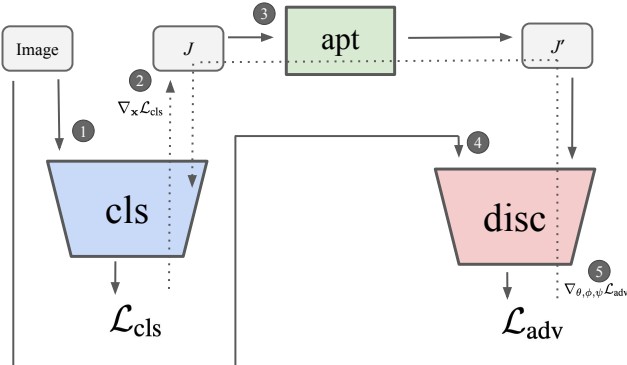

Figure 1: Training architecture of JARN.

---

**Algorithm 1:** Jacobian Adversarially Regularized Network

1  **Input:** Training data $\mathcal{D}_{\text{train}}$, Learning rates for classifier $f_{\text{cls}}$, adaptor $f_{\text{apt}}$ and discriminator $f_{\text{disc}}$: $(\alpha, \beta, \gamma)$
2  **for** *each training iteration* **do**
3       Sample $(\mathbf{x}, \mathbf{y}) \sim \mathcal{D}_{\text{train}}$
4       $\mathbf{x} \leftarrow \mathbf{x} + \xi, \quad \xi_i \sim \text{unif}[-\varepsilon, \varepsilon]$
5       $\mathcal{L}_{\text{cls}} \leftarrow -\mathbf{y}^{\top} \log f_{\text{cls}}(\mathbf{x})$            ▷ (1) Compute classification cross-entropy loss
6       $J \leftarrow \nabla_{\mathbf{x}} \mathcal{L}_{\text{cls}}$            ▷ (2) Compute Jacobian matrix
7       $J' \leftarrow f_{\text{apt}}(J)$            ▷ (3) Adapt Jacobian to image domain
8       $\mathcal{L}_{\text{adv}} \leftarrow \log f_{\text{disc}}(\mathbf{x}) + \log(1 - f_{\text{disc}}(J'))$            ▷ (4) Compute adversarial loss
9       $\theta \leftarrow \theta - \alpha \nabla_{\theta}(\mathcal{L}_{cls} + \lambda_{adv} \mathcal{L}_{adv})$            ▷ (5a) Update the classifier $f_{\text{cls}}$ to minimize $\mathcal{L}_{cls}$ and $\mathcal{L}_{adv}$
10       $\psi \leftarrow \psi - \beta \nabla_{\psi} \mathcal{L}_{adv}$            ▷ (5b) Update the adaptor $f_{\text{apt}}$ to minimize $\mathcal{L}_{adv}$
11       $\phi \leftarrow \phi + \gamma \nabla_{\phi} \mathcal{L}_{adv}$            ▷ (5c) Update the discriminator $f_{\text{disc}}$ to maximize $\mathcal{L}_{adv}$

---

### 3.1 THEORETICAL ANALYSIS

Here, we study the link between JARN's adversarial regularization term with the notion of linearized robustness. Assuming a non-parameteric setting where the models have infinite capacity, we have the following theorem while optimizing $G$ with the adversarial loss $\mathcal{L}_{adv}$.

**Theorem 3.1.** *The global minimum of $\mathcal{L}_{adv}$ is achieved when $G(\mathbf{x})$ maps $\mathbf{x}$ to itself, i.e., $G(\mathbf{x}) = \mathbf{x}$.*

Its proof is deferred to § A. If we assume Jacobian $J$ of our classifier $f_{\text{cls}}$ to be the direct output of $G$, then $J = G(\mathbf{x}) = \mathbf{x}$ at the global minimum of the adversarial objective.

In Etmann et al. (2019), it is shown that the linearized robustness of a model is loosely upper-bounded by the alignment between the Jacobian and the input image. More concretely, denoting $\Psi^i$ as the logits value of class $i$ in a classifier $F$, its linearized robustness $\rho$ can be expressed as $\rho(\mathbf{x}) := \min_{j \neq i^*} \frac{\Psi^{i^*}(\mathbf{x}) - \Psi^j(\mathbf{x})}{\|\nabla_{\mathbf{x}} \Psi^{i^*}(\mathbf{x}) - \nabla_{\mathbf{x}} \Psi^j(\mathbf{x})\|}$. Here we quote the theorem from Etmann et al. (2019):

**Theorem 3.2** (Linearized Robustness Bound). *(Etmann et al., 2019) Defining $i^* = \arg\max_i \Psi^i$ and $j^* = \arg\max_{j \neq i^*} \Psi^j$ as top two prediction, we let the Jacobian with respect to the difference in top two logits be $g := \nabla_{\mathbf{x}}(\Psi^{i^*} - \Psi^{j^*})(\mathbf{x})$. Expressing alignment between the Jacobian with the input as $\alpha(\mathbf{x}) = \frac{|\langle \mathbf{x}, g \rangle|}{\|g\|}$, then*

$$\rho(\mathbf{x}) \leq \alpha(\mathbf{x}) + \frac{C}{\|g\|} \tag{13}$$

*where $C$ is a positive constant.*

Combining with what we have in Theorem 3.1, assuming $J$ to be close to $g$ in a fixed constant term, the alignment term $\alpha(\mathbf{x})$ in Equation (13) is maximum when $\mathcal{L}_{\text{adv}}$ reaches its global minimum. Though this is not a strict upper bound and, to facilitate the training in JARN in practice, we use an adaptor network to transform the Jacobian, i.e., $J' = f_{\text{apt}}(J)$, our experiments show that model robustness can be improved with this adversarial regularization.

## 4 EXPERIMENTS

We conduct experiments on three image datasets, MNIST, SVHN and CIFAR-10 to evaluate the adversarial robustness of models trained by JARN.

### 4.1 MNIST

**Setup**  MNIST consists of 60k training and 10k test binary-colored images. We train a CNN, sequentially composed of 3 convolutional layers and 1 final softmax layer. All 3 convolutional layers have a stride of 5 while each layer has an increasing number of output channels (64-128-256). For JARN, we use $\lambda_{\text{adv}} = 1$, a discriminator network of 2 CNN layers (64-128 output channels) and update it for every 10 $f_{\text{cls}}$ training iterations. We evaluate trained models against adversarial examples with $l_\infty$ perturbation $\varepsilon = 0.3$, crafted from FGSM and PGD (5 & 40 iterations). FGSM generates weaker adversarial examples with only one gradient step and is weaker than the iterative PGD method.

**Results**  The CNN trained with JARN shows improved adversarial robustness from a standard model across the three types of adversarial examples (Table 1). In the MNIST experiments, we find that data augmentation with uniform noise to pixels alone provides no benefit in robustness from the baseline.

Table 1: MNIST accuracy (%) on adversarial and clean test samples.

| Model | FGSM | PGD5 | PGD40 | Clean |
|---|---|---|---|---|
| Standard | 76.5 | 0 | 0 | 98.7 |
| Uniform Noise | 77.5 | 0 | 0.02 | 98.7 |
| JARN | **98.4** | **98.1** | **98.1** | 98.8 |

### 4.2 SVHN

**Setup**  SVHN is a 10-class house number image classification dataset with 73257 training and 26032 test images, each of size $32 \times 32 \times 3$. We train the Wide-Resnet model following hyperparameters from (Madry et al., 2017)'s setup for their CIFAR-10 experiments. For JARN, we use $\lambda_{\text{adv}} = 5$, a discriminator network of 5 CNN layers (16-32-64-128-256 output channels) and update it for every 20 $f_{\text{cls}}$ training iterations. We evaluate trained models against adversarial examples with ($\varepsilon = 8/255$), crafted from FGSM and 5, 10, 20-iteration PGD attack.

**Results**  Similar to the findings in § 4.1, JARN advances the adversarial robustness of the classifier from the standard baseline against all four types of attacks. Interestingly, uniform noise image augmentation increases adversarial robustness from the baseline in the SVHN experiments, concurring with previous work that shows noise augmentation improves robustness (Ford et al., 2019).

Table 2: SVHN accuracy (%) on adversarial and clean test samples.

| Model | FGSM | PGD5 | PGD10 | PGD20 | Clean |
|---|---|---|---|---|---|
| Standard | 64.4 | 26.0 | 5.47 | 1.96 | 94.7 |
| Uniform Noise | 65.0 | 42.6 | 18.4 | 9.21 | 95.3 |
| JARN | **67.2** | **57.5** | **37.7** | **26.79** | 94.9 |

## 4.3 CIFAR-10

**Setup** CIFAR-10 contains $32 \times 32 \times 3$ colored images labeled as 10 classes, with 50k training and 10k test images. We train the Wide-Resnet model using similar hyperparameters to (Madry et al., 2017) for our experiments. Following the settings from Madry et al. (2017), we compare with a strong adversarial training baseline (PGD-AT7) that involves training the model with adversarial examples generate with 7-iteration PGD attack. For JARN, we use $\lambda_{\text{adv}} = 1$, a discriminator network of 5 CNN layers (32-64-128-256-512 output channels) and update it for every 20 $f_{\text{cls}}$ training iterations. We evaluate trained models against adversarial examples with ($\varepsilon = 8/255$), crafted from FGSM and PGD (5, 10 & 20 iterations). We also add in a fast gradient sign attack baseline (FGSM-AT1) that generates adversarial training examples with only 1 gradient step. Though FGSM-trained models are known to rely on obfuscated gradients to counter weak attacks, we augment it with JARN to study if there is additive robustness benefit against strong attacks. We also implemented double backpropagation (Drucker & Le Cun, 1991; Ross & Doshi-Velez, 2018) to compare.

**Results** Similar to results from the previous two datasets, the JARN classifier performs better than the standard baseline for all four types of adversarial examples. Compared to the model trained with uniform-noise augmentation, JARN performs closely in the weaker FGSM attack while being more robust against the two stronger PGD attacks. JARN also outperforms the double backpropagation baseline, showing that regularizing for salient Jacobians confers more robustness than regularizing for smaller Jacobian Frobenius norm values. The strong PGD-AT7 baseline shows higher robustness against PGD attacks than the JARN model. When we train JARN together with 1-step adversarial training (JARN-AT1), we find that the model's robustness exceeds that of strong PGD-AT7 baseline on all four adversarial attacks, suggesting JARN's gain in robustness is additive to that of AT.

Table 3: CIFAR-10 accuracy (%) on adversarial and clean test samples.

| Model | FGSM | PGD5 | PGD10 | PGD20 | Clean |
|---|---|---|---|---|---|
| Standard | 13.4 | 0 | 0 | 0 | 95.0 |
| Uniform Noise | 67.4 | 44.6 | 19.7 | 7.48 | 94.0 |
| FGSM-AT1 | **94.5** | 0.25 | 0.02 | 0.01 | 91.7 |
| Double Backprop | 28.3 | 0.05 | 0 | 0 | 95.7 |
| JARN | 67.2 | 50.0 | 27.6 | 15.5 | 93.9 |
| PGD-AT7 | 56.2 | 55.5 | 47.3 | 45.9 | 87.3 |
| JARN-AT1 | 65.7 | **60.1** | **51.8** | **46.7** | 84.8 |

### 4.3.1 GENERALIZATION OF ROBUSTNESS

Adversarial training (AT) based defenses generally train the model on examples generated by perturbation of a fixed $\varepsilon$. Unlike AT, JARN by itself does not have $\varepsilon$ as a training parameter. To study how JARN-AT1 robustness generalizes, we conduct PGD attacks of varying $\varepsilon$ and strength (5, 10 and 20 iterations). We also include another PGD-AT7 baseline that was trained at a higher $\varepsilon = (12/255)$. JARN-AT1 shows higher robustness than the two PGD-AT7 baselines against attacks with higher $\varepsilon$ values ($\leq 8/255$) across the three PGD attacks, as shown in Figure 2. We also observe that the PGD-AT7 variants outperform each other on attacks with $\varepsilon$ values close to their training $\varepsilon$, suggesting that their robustness is more adapted to resist adversarial examples that they are trained on. This relates to findings by Tramèr & Boneh (2019) which shows that robustness from adversarial training is highest against the perturbation type that models are trained on.

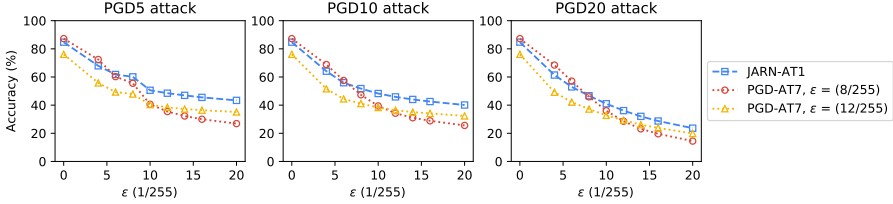

Figure 2: Generalization of model robustness to PGD attacks of different $\varepsilon$ values.

### 4.3.2 LOSS LANDSCAPE

We compute the classification loss value along the adversarial perturbation's direction and a random orthogonal direction to analyze the loss landscape of the models. From Figure 3, we see that the models trained by the standard and FGSM-AT method display loss surfaces that are jagged and non-linear. This explains why the FGSM-AT display modest accuracy at the weaker FGSM attacks but fail at attacks with more iterations, a phenomenon called obfuscated gradients (Carlini & Wagner, 2017; Uesato et al., 2018) where the initial gradient steps are still trapped within the locality of the input but eventually escape with more iterations. The JARN model displays a loss landscape that is less steep compared to the standard and FGSM-AT models, marked by the much lower (1 order of magnitude) loss value in Figure 3c. When JARN is combined with one iteration of adversarial training, the JARN-AT1 model is observed to have much smoother loss landscapes, similar to that of the PGD-AT7 model, a strong baseline previously observed to be free of obfuscated gradients. This suggests that JARN and AT have additive benefits and JARN-AT1's adversarial robustness is not attributed to obfuscated gradients.

A possible explanation behind the improved robustness through increasing Jacobian saliency is that the space of Jacobian shrinks under this regularization, i.e., Jacobians have to resemble non-noisy images. Intuitively, this means that there would be fewer paths for an adversarial example to reach an optimum in the loss landscape, improving the model's robustness.

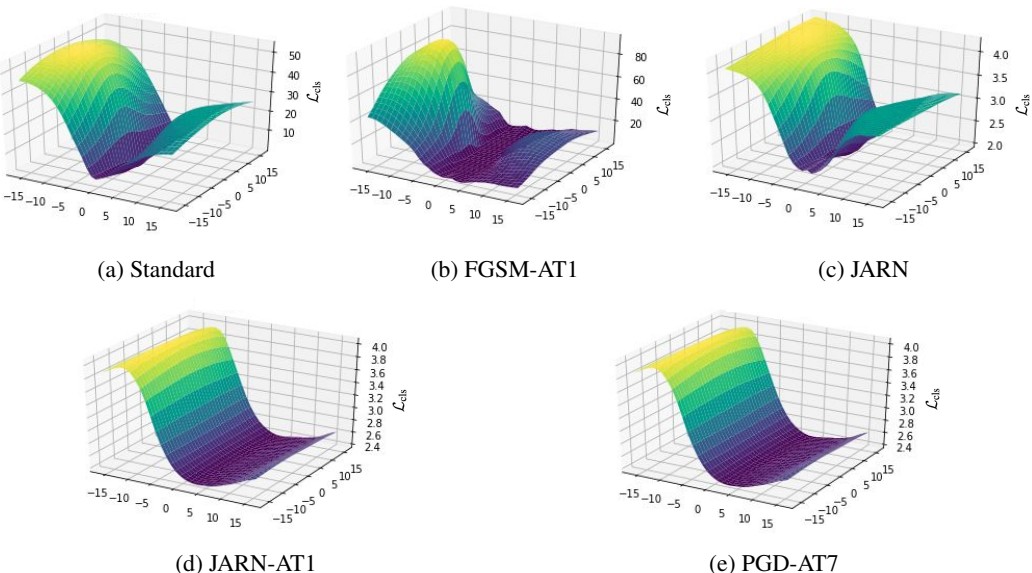

Figure 3: Loss surfaces of models along the adversarial perturbation and a random direction.

### 4.3.3 SALIENCY OF JACOBIAN

The Jacobian matrices of JARN model and PGD-AT are salient and visually resemble the images more than those from the standard model (Figure 4). Upon closer inspection, the Jacobian matrices of the PGD-AT model concentrate their values at small regions around the object of interest whereas those of the JARN model cover a larger proportion of the images. One explanation is that the JARN model is trained to fool the discriminator network and hence generates Jacobian that contains details of input images to more closely resemble them.

### 4.3.4 COMPUTE TIME

Training with JARN is computationally more efficient when compared to adversarial training (Table 4). Even when combined with FGSM adversarial training JARN, it takes less than half the time of 7-step PGD adversarial training while outperforming it in robustness.

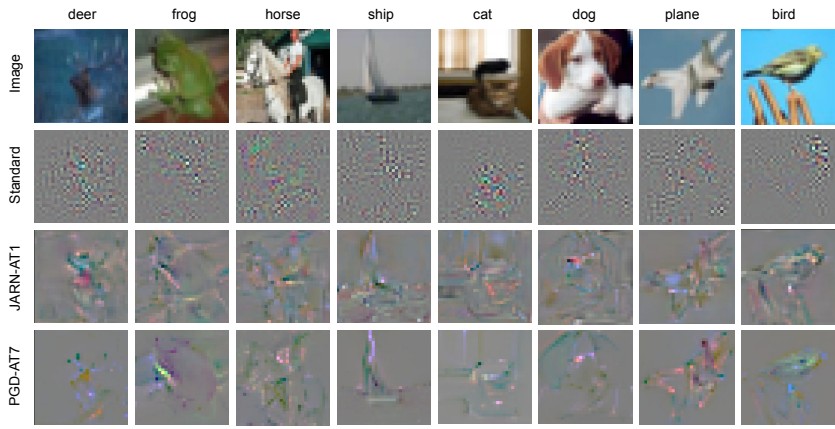

Figure 4: Jacobian matrices of CIFAR-10 models.

Table 4: Average wall-clock time per training epoch for CIFAR-10 adversarial defenses.

| Model | PGD-AT7 | JARN-AT1 | FGSM-AT1 | JARN only |
|---|---|---|---|---|
| Time (sec) | 704 | 294 | 267 | 217 |

### 4.3.5 SENSITIVITY TO HYPERPARAMETERS

The performance of GANs in image generation has been well-known to be sensitive to training hyperparameters. We test JARN performance across a range of $\lambda_{adv}$, batch size and discriminator update intervals that are different from § 4.3 and find that its performance is relatively stable across hyperparameter changes, as shown in Appendix Figure 5. In a typical GAN framework, each training step involves a real image sample and an image generated from noise that is decoupled from the real sample. In contrast, a Jacobian is conditioned on its original input image and both are used in the same training step of JARN. This training step resembles that of VAE-GAN (Larsen et al., 2015) where pairs of real images and its reconstructed versions are used for training together, resulting in generally more stable gradients and convergence than GAN. We believe that this similarity favors JARN's stability over a wider range of hyperparameters.

### 4.3.6 BLACK-BOX TRANSFER ATTACKS

Transfer attacks are adversarial examples generated from an alternative, substitute model and evaluated on the defense to test for gradient masking (Papernot et al., 2016; Carlini et al., 2019). More specifically, defenses relying on gradient masking will display lower robustness towards transfer attacks than white-box attacks. When evaluated on such black-box attacks using adversarial examples generated from a PGD-AT7 trained model and their differently initialized versions, both JARN and JARN-AT1 display higher accuracy than when under white-box attacks (Table 5). This demonstrates that JARN's robustness does not rely on gradient masking. Rather unexpectedly, JARN performs better than JARN-AT1 under the PGD-AT7 transfer attacks, which we believe is attributed to its better performance on clean test samples.

Table 5: CIFAR-10 accuracy (%) on transfer attack where adversarial examples are generated from a PGD-AT7 trained model.

| Model | PGD-AT7 | | Same Model | | White-box | | Clean |
|---|---|---|---|---|---|---|---|
| | FGSM | PGD20 | FGSM | PGD20 | FGSM | PGD20 | |
| JARN | 79.6 | 76.7 | 73.6 | 17.4 | 67.2 | 15.5 | 93.9 |
| JARN-AT1 | 66.4 | 63.0 | 70.3 | 59.3 | 65.7 | 46.7 | 84.8 |

## 5 CONCLUSIONS

In this paper, we show that training classifiers to give more salient input Jacobian matrices that resemble images can advance their robustness against adversarial examples. We achieve this through an adversarial regularization framework (JARN) that train the model's Jacobians to fool a discriminator network into classifying them as images. Through our experiments in three image datasets, JARN boosts adversarial robustness of standard models and give competitive performance when added on to weak defenses like FGSM. Our findings open the viability of improving the saliency of Jacobian as a new avenue to boost adversarial robustness.

### ACKNOWLEDGMENTS

This work is funded by the National Research Foundation, Singapore under its AI Singapore programme [Award No.: AISG-RP-2018-004] and the Data Science and Artificial Intelligence Research Center (DSAIR) at Nanyang Technological University.

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

## A  PROOF OF THEOREM 3.1

**Theorem A.1.** *The global minimum of $\mathcal{L}_{adv}$ is achieved when $G(\mathbf{x})$ maps $\mathbf{x}$ to itself, i.e., $G(\mathbf{x}) = \mathbf{x}$.*

*Proof.* From (Goodfellow et al., 2014), for a fixed $G$, the optimal discriminator is

$$f_{\text{disc}}^*(\mathbf{x}) = \frac{p_{\text{data}}(\mathbf{x})}{p_{\text{data}}(\mathbf{x}) + p_G(\mathbf{x})} \tag{14}$$

We can include the optimal discriminator into Equation (9) to get

$$
\begin{aligned}
\mathcal{L}_{\text{adv}}(G) &= \mathbb{E}_{\mathbf{x} \sim p_{\text{data}}}[\log f_{\text{disc}}^*(\mathbf{x})] + \mathbb{E}_{\mathbf{x} \sim p_{\text{data}}}[\log(1 - f_{\text{disc}}^*(G(\mathbf{x})))] \\
&= \mathbb{E}_{\mathbf{x} \sim p_{\text{data}}}[\log f_{\text{disc}}^*(\mathbf{x})] + \mathbb{E}_{\mathbf{x} \sim p_G}[\log(1 - f_{\text{disc}}^*(\mathbf{x}))] \\
&= \mathbb{E}_{\mathbf{x} \sim p_{\text{data}}}\left[\log \frac{p_{\text{data}}(\mathbf{x})}{p_{\text{data}}(\mathbf{x}) + p_G(\mathbf{x})}\right] + \mathbb{E}_{\mathbf{x} \sim p_G}\left[\log \frac{p_G(\mathbf{x})}{p_{\text{data}}(\mathbf{x}) + p_G(\mathbf{x})}\right] \\
&= \mathbb{E}_{\mathbf{x} \sim p_{\text{data}}}\left[\log \frac{p_{\text{data}}(\mathbf{x})}{\frac{1}{2}(p_{\text{data}}(\mathbf{x}) + p_G(\mathbf{x}))}\right] + \mathbb{E}_{\mathbf{x} \sim p_G}\left[\log \frac{p_G(\mathbf{x})}{\frac{1}{2}(p_{\text{data}}(\mathbf{x}) + p_G(\mathbf{x}))}\right] - 2\log 2 \\
&= KL\left(p_{\text{data}} \,\Big\|\, \frac{p_{\text{data}} + p_G}{2}\right) + KL\left(p_G \,\Big\|\, \frac{p_{\text{data}} + p_G}{2}\right) - \log 4 \\
&= 2 \cdot JS(p_{\text{data}} \| p_G) - \log 4
\end{aligned}
\tag{15}
$$

where $KL$ and $JS$ are the Kullback-Leibler and Jensen-Shannon divergence respectively. Since the Jensen-Shannon divergence is always non-negative, $\mathcal{L}_{\text{adv}}(G)$ reaches its global minimum value of $-\log 4$ when $JS(p_{\text{data}} \| p_G) = 0$. When $G(\mathbf{x}) = \mathbf{x}$, we get $p_{\text{data}} = p_G$ and consequently $JS(p_{\text{data}} \| p_G) = 0$, thus completing the proof.

$\square$

## B  SENSITIVITY TO HYPERPARAMETERS

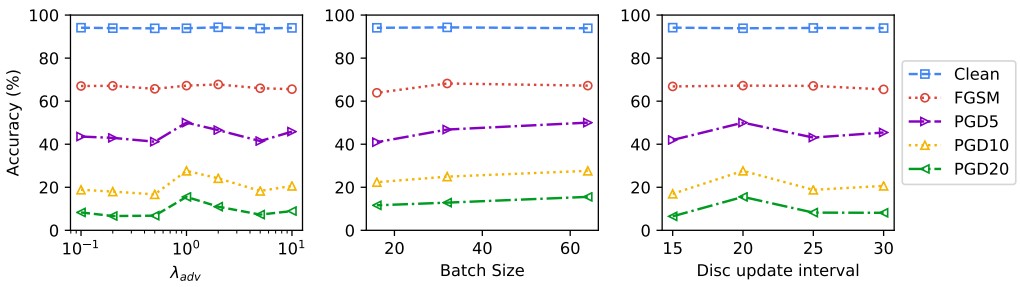

Figure 5: Accuracy of JARN with different hyperparameters on CIFAR-10 test samples.

