# OpenReview forum: "Jacobian Adversarially Regularized Networks for Robustness"
_ICLR.cc/2020/Conference — Accept (Poster)_

### Official Review · AnonReviewer2 · 2019-10-18
**Official Blind Review #2**

**Rating:** 3

**Review:**

I think the main contribution of this paper is that it introduces a new way of robust training by encouraging Jacobian saliency. Previous research like Etmann et al 2019 and Tsipras et al 2018 showed that robustness leads to saliency. But surprisingly, this paper shows the other way, saliency map can also lead to robustness, which indicates a stronger connection between these two. In general, I like the intuition behind this paper, since it introduces a new perspective of robust training.

The training method proposed in this paper is still kind of preliminary, though. I suspect that training a GAN together with the classifier will cost even more time than min-max adversarial training or some certified robust training methods. It would be great if the authors can provide the training time comparison between JARN and some state-of-the-art robust training methods. Another concern is reproducibility since the training process of GAN is sensitive to hyperparameter selection. It would be better if the author can have some discussion on the training process to show that the reported performance of the defense is easy to reproduce instead of cherry-picking.  Also, there are too many works on robustness defense that have been proven ineffective (consider the works by Carlini). Since this is a completely new way of robust training and there is no certified guarantee, I suggest the authors refer [1] to evaluate the effectiveness of the defense more thoroughly to convince the readers that it really works. Especially, evaluation under adaptive attack is necessary.

I think this is a very interesting work. But since this method is completely new, more detailed information is needed to convince me that it really works. If it does work, I believe there must exist better ways to encourage Jacobian saliency than using a GAN.

[1] Carlini, Nicholas, et al. "On evaluating adversarial robustness." arXiv preprint arXiv:1902.06705 (2019).

**Experience Assessment:**

I have published one or two papers in this area.

**Review Assessment: Checking Correctness Of Derivations And Theory:**

I assessed the sensibility of the derivations and theory.

**Review Assessment: Checking Correctness Of Experiments:**

I assessed the sensibility of the experiments.

**Review Assessment: Thoroughness In Paper Reading:**

I read the paper thoroughly.

---

> ### Author Response · Authors · 2019-11-15
> **Response to Review #2**
>
> We thank the reviewer for the constructive comments and would like to respond to them as follows,
>
> > Computational cost of JARN:
> Following the advice of the reviewer, we have added experiments to compare JARN’s computational efficiency with PGD adversarial training in Section 4.3.4 and Table 4. Even when combined with 1-step adversarial training, JARN takes less than half the time compared to 7-step PGD adversarial training while outperforming it. JARN’s GAN component adds a relatively small computational burden since the discriminator only updates once every 20 classifier update steps in the CIFAR-10 experiments and is much smaller in parameter size compared to the main classifier. Moreover, we find that using JARN framework only on the last few epoch (25%) to train the classifier confers similar adversarial robustness compared to training with JARN for the whole duration, further increasing its efficiency.
>
>
> > Reproducibility and sensitivity to hyperparameters:
> To address the reviewer’s comment, we have added more experiments to test JARN performance across a range of key hyperparameters ($\lambda_{adv}$, batch size and discriminator update intervals) that are different from Section 4.3 and find that its performance is relatively stable across key hyperparameter changes (Section 4.3.5 and Appendix Figure 5).
>
> In a typical GAN framework, each training step involves a real image sample and an image generated from noise that is decoupled from the real sample. In contrast, a Jacobian is conditioned on its original input image and both are used in the same training step of JARN. This training step resembles that of VAE-GAN [1] where pairs of real images and its reconstructed versions are used for training together, resulting in generally more stable gradients and convergence than GAN. We believe that this similarity favors JARN's stability over a wider range of hyperparameters. To further ensure reproducibility, we will release the source code after the paper’s acceptance.
>
>
> > More evaluation of the defense:
> Following the reviewer’s suggestion to evaluate the JARN classifier’s robustness more comprehensively [2], we have added experiments of black-box transfer attacks on JARN (Section 4.3.6 and Table 5). Defenses relying on gradient masking will display lower robustness towards transfer attacks than white-box attacks [2,3]. When evaluated on such black-box attacks using adversarial examples generated from a PGD-AT7 trained model and their differently initialized version, both JARN and JARN-AT1 display higher accuracy than when under white-box attacks (Table 5), a sign that JARN's robustness does not rely on gradient masking. We would like to also point out Section 4.3.1 where evaluation is carried out on attacks over a range of different parameters to test the generalization of JARN’s robustness.
>
>
> > Other ways to encourage Jacobian saliency:
> Since our main research question is to study if regularizing for image-resembling Jacobians can lead to robustness, we wish to study this by directly training for generated Jacobian to look like images. Due to GAN’s remarkable success in synthetic image generation, we incorporate GAN into the JARN framework for this objective. While we agree that other ways that can encourage Jacobian saliency are interesting future work, we would like to point out that this paper is the first to show this approach can lead to improved robustness.
>
>
> [1] Larsen et al., 2015 “Autoencoding beyond pixels using a learned similarity metric.” arXiv preprint arXiv:1512.09300, 2015
>
> [2] Carlini, Nicholas, et al. "On evaluating adversarial robustness." arXiv preprint arXiv:1902.06705, 2019.
>
> [3] Papernot et al., 2015 “Transferability in machine learning: from phenomena to black-box attacks using adversarial samples.” arXiv preprint arXiv:1605.07277, 2016

---

### Official Review · AnonReviewer1 · 2019-10-22
**Official Blind Review #1**

**Rating:** 6

**Review:**

This paper proposes a novel regularization strategy for improving the robustness of networks to adversarial noise. A term is added to the standard supervised cross-entropy loss that encourages the Jacobian of the network to itself be interpreted as a valid image. This "regularization" term is constructed by running the input-output Jacobian of the classification network through an "adapter network" and then in turn interpreting its output as a "generator" in a GAN setup. A separate discriminator network is training to distinguish real input images from these adapter-processed input-output Jacobians. The overall regularization is the standard minimax GAN loss applied to this generator/discriminator setup. The impetus for this stems from a previous observation that salient or interpretable input-output Jacobians naturally arise for networks that have undergone adversarial training to increase robustness.

Although this whole setup seems to be a little "Rube-Goldberg"-esque, I think there's some real sensible reasons for this sort of regularization to make intuitive sense. The input-output Jacobian characterizes how much the output (i.e. the logits) are affected by small changes to the input. The Jacobian, reinterpreted as living in the input image space (as the authors do), is a map of which input pixels have the strongest effect on the output of the network. If the Jacobian image looks like the underlying input image -- in particular, highlighting the labeled object -- this indicates that changing those pixels will result in the largest change on the network output. (This should be clear when looking at Figure 4 of the paper.) On the other hand, adversarial noise by definition leaves the underlying object alone (so that a human isn't aware of the perturbation) and modifies other pixels. Models that fall for such adversarial noise will not have salient Jacobians.

This is an amusing original idea, and I think this paper probably should be accepted to ICLR -- though I don't hold that position very strongly. However, I think the most interesting point is idea of Jacobian saliency, which is from prior work (Tsipras et al., 2018) that I haven't read. Therefore, I'm not sure how significant this paper is on it's own. Regardless, I would have liked to see more discussion in the paper of why Jacobian saliency should confer robustness (as I tried to do in the paragraph above), with perhaps some additional experiments designed around understanding whether this intuition (or something similar) is actually correct. There's some discussion of the theory behind the method in section 3.1, but it's not very intuitive to the situation at hand (non-linear neural networks), and I don't find it particularly informative.

Finally, some effort is spent arguing that this method is more computational efficient than adversarial training -- I wonder if that's still true when the all the complexity of GAN training is taken into account or how to consider that point when part of the conclusion is that their method is best when it is also combined with some amount of adversarial training.

**Experience Assessment:**

I have published one or two papers in this area.

**Review Assessment: Checking Correctness Of Derivations And Theory:**

I assessed the sensibility of the derivations and theory.

**Review Assessment: Checking Correctness Of Experiments:**

I assessed the sensibility of the experiments.

**Review Assessment: Thoroughness In Paper Reading:**

I read the paper at least twice and used my best judgement in assessing the paper.

---

> ### Author Response · Authors · 2019-11-15
> **Response to Review #1**
>
> We thank the reviewer for the thoughtful comments and the interest in our work. Our detailed response to the questions follows.
>
> > The idea of Jacobian saliency from prior work:
> In contrast to prior work, to the best of our knowledge, our paper is the first with the aim to train the classifier’s Jacobian to resemble input images more closely as a way to improve robustness. We summarize those previous studies here to better contrast with our paper:
>
> The focus of the work from (Tsipras et al., 2018) is on the trade-off between a model’s standard accuracy and its robustness while showing that robust models learn different feature representations than their non-robust counterparts. The authors observed from their experiments that the adversarial examples generated from robust models look perceptually different from the original images. They attributed this to the higher saliency of the Jacobian generated at each gradient step of the adversarial attack on robust models. While (Tsipras et al., 2018) is one of the first to point out this observation, the authors did not propose a new defense based on it.
>
> (Etmann et al., 2019) further investigated this observation by proving that linearized robustness (distance from samples to decision boundary) increases as the alignment (unit vector cosine similarity) between the Jacobian and input image grows in linear models. The authors train robust models using double backpropagation (Drucker & Le Cun, 1992) and show empirically the relationship between robustness and alignment weakens for non-linear models but did not propose a new defense in the paper.
>
> (Drucker & Le Cun, 1991; Ross & Doshi-Velez, 2018; Jakubovitz & Giryes, 2018; Hoffman et al., 2019; Simon-Gabriel et al., 2019) use a regularization term to reduce the Jacobian's Frobenius norm. While their approaches constrain the effect of individual pixels’ perturbation on the model prediction, our approach focuses on the intuition to improve robustness by training for salient Jacobians that look like images. Our newly added experiments (Section 4.3 and Table 3) show that JARN outperforms this earlier approach in adversarial robustness.
>
>
> > Discussion on why Jacobian saliency confers robustness:
> We thank the reviewer for the insightful discussion and would like to share our interpretation. The input Jacobian indeed characterizes how the final logits are affected by small changes to the pixels. Indeed, regularizing the Jacobian to resemble the input would likely result in adversarial perturbations that perceptually change the labeled object. A possible explanation behind the improved robustness through increasing Jacobian saliency is that the space of input-output Jacobian shrinks under this regularization, i.e. Jacobians have to resemble non-noisy images. Intuitively, this means that there would be fewer paths for an adversarial example to reach an optimum in the loss landscape, improving the model’s robustness. As suggested by the reviewer, we added more discussion on this in Section 4.3.2.
>
>
> > Computational efficiency of JARN compared to adversarial training:
> Following the suggestion of the reviewer, we have added experiments to compare JARN’s computational efficiency with adversarial training in Section 4.3.4 and Table 4. Even when combined with 1-step adversarial training, JARN takes less than half the time compared to 7-step PGD adversarial training while outperforming it. In our experiments on CIFAR-10, the JARN discriminator only updates once every 20 classifier update steps and is much smaller in parameter size compared to the main classifier, explaining JARN’s efficiency. Moreover, we find that using JARN framework only on the last few epoch (25%) to train the classifier confers similar adversarial robustness compared to training with JARN for the whole duration.
>
> References:
> - Tsipras et al., “Robustness may be at odds with accuracy.” arXiv preprint arXiv:1805.12152, 2018.
>
> - Etmann et al., “On the connection between adversarial robustness and saliency map interpretability.” arXiv preprint arXiv:1905.04172, 2019.
>
> -  Drucker and Le Cun "Double backpropagation increasing generalization performance" IEEE Transactions on Neural Networks, 3(6):991–997, 1992.
>
> - Ross & Doshi-Velez, “Improving the adversarial robustness and interpretability of deep neural networks by regularizing their input gradients.” In Thirty-second AAAI conference on artificial intelligence, 2018.
>
> - Jakubovitz & Giryes, “Improving dnn robustness to adversarial attacks using jacobian regularization” In Proceedings of the European Conference on Computer Vision (ECCV), pp.
> 514–529, 2018.
>
> - Hoffman et al., “Robust learning with jacobian regularization.” arXiv preprint arXiv:1908.02729, 2019.
>
> - Simon-Gabriel et al., "First-order Adversarial Vulnerability of Neural Networks and Input Dimension" In International Conference on Machine Learning, pp. 5809–5817, 2019

---

### Official Review · AnonReviewer3 · 2019-10-23
**Official Blind Review #3**

**Rating:** 6

**Review:**

Summary:
It was previously observed that models that were more robust to adversarial perturbation had more interpretable jacobian. The authors attempt to train for interpretable jacobian in order to improve the robustness of the model.
This is done by employing a GAN-like procedure where a discriminator attempts to distinguish between the transformed jacobian matrix (fake images, equivalent to generator) and real images.

Experiments indicates that this improves robustness compared to unprotected models and approximately similarly to models trained with adversarial training.

Comments:
* The motivation given for this line of research is the cost of adversarial training (2nd paragraph of Section 3)
No experimental comparison is given with regards to the time it takes to train a model with adversarial training, versus the time it takes to train a model with JARN. It is also important to note that this introduces additional complexity (needs to choose an architecture for the discriminator, tune proper learning rates, etc...), which is not mentionned.

* Why not test simpler jacobian regularization method as proposed by other papers (see below). Proposition 3 of Simon-Gabriel et al. shows that results similar to adversarial training can be obtained, and they don't need several iterations like adversarial training, nor do they need to train an additional discriminator like your method.

Opinion:
The paper provides an interesting proof of concept for a method, showing that it is feasible. It however doesn't make the the case for why it is a good idea. Discussion and comparison to very significant related work is missing and experimental measurement of any advantages of the proposed method vs. adversarial training is lacking. I think that these aspects should be improved before the paper is ready for publication.

Typos:
Line 11 in Algorithm 1 -> The label is wrong, i assume it's "Update the discriminator f_disc to maximize L_adv"

Related works that needs discussing:
- Drucker, Lecun 91, "Double backpropagation increasing generalization performance" for other regularizer on the jacobian, discusses generalization rather than robustness.
- Simon-Gabriel et al., "First-order Adversarial Vulnerability of Neural Networks and Input Dimension"


**Experience Assessment:**

I have read many papers in this area.

**Review Assessment: Checking Correctness Of Derivations And Theory:**

I assessed the sensibility of the derivations and theory.

**Review Assessment: Checking Correctness Of Experiments:**

I assessed the sensibility of the experiments.

**Review Assessment: Thoroughness In Paper Reading:**

I read the paper thoroughly.

---

> ### Author Response · Authors · 2019-11-15
> **Response to Review #3**
>
> We thank the reviewer for the helpful comments and would like to respond to them as follows,
>
> > Cost of JARN training:
> To address the comment on computational cost, we have added experiments to compare JARN’s computational efficiency with PGD adversarial training in Section 4.3.4 and Table 4. Even when combined with 1-step adversarial training, JARN takes less than half the time compared to 7-step PGD adversarial training while outperforming it. JARN’s GAN component adds a relatively small computational burden since the discriminator only updates once every 20 classifier update steps in the CIFAR-10 experiments and is much smaller in parameter size compared to the main classifier. Moreover, we find that using JARN framework only on the last few epoch (25%) to train the classifier confers similar adversarial robustness compared to training with JARN for the whole duration, further supporting its efficiency.
>
> Regarding the concern of hyperparameter tuning, we have added more experiments to test JARN performance across a range of key hyperparameters ($\lambda_{adv}$, batch size and discriminator update intervals) that are different from Section 4.3 and find that its performance is relatively stable across hyperparameter changes (Appendix Figure 5). In a typical GAN framework, each training step involves a real image sample and an image generated from noise that is decoupled from the real sample. In contrast, a Jacobian is conditioned on its original input image and both are used in the same training step of JARN. This training step resembles that of VAE-GAN [1] where pairs of real images and its reconstructed versions are used for training together, resulting in generally more stable gradients and convergence than GAN. We believe that this similarity favors JARN's stability over a wider range of hyperparameters.
>
>
> > Why not test other Jacobian regularization methods?
> Following the suggestion of the reviewer, we implemented double backpropagation (DBP) [2-4] as a baseline with additional results to compare in our paper (Section 4.3 and Table 3). We found that DBP provides robustness against FGSM attacks compared to standard training but is outperformed by JARN across all the attacks. While Proposition 3 in [5] shows that DBP is equivalent to training with $l_2$ adversarial examples, we believe they are single-step adversarial examples rather than the stronger multi-step adversarial examples generated by iterative methods like PGD. This explains DBP’s performance under the more recent and stronger PGD attacks. The comparison demonstrates that JARN’s approach to robustness through the saliency of Jacobian is fundamentally different and more effective than DBP’s approach through minimizing Jacobian's Frobenius norm. We further discuss the difference from the prior art in the following paragraphs for completeness.
>
> Since the contribution of our paper is to study if regularizing for image-resembling Jacobians can be a new way to improve robustness, we wish to study this by directly training for generated Jacobians to look like images. [2-4] employ a regularization term to minimize the Jacobian's Frobenius norm together with the standard training objective. While their approaches improve robustness by reducing the effect that perturbations in individual pixel have on the classifier’s prediction through the Jacobians’ norm, it does not have the aim to optimize Jacobians to explicitly resemble their corresponding images. We describe those studies in more detail here and in the revision (Section 2: Non-Adversarial Training Regularization):
>
> [2] first proposed this to improve model generalization on natural test samples and called it ‘double backpropagation’ while [3,4] are concurrent studies that evaluate this method against adversarial examples. [5] proved that double backpropagation is equivalent to adversarial training with $l_2$ examples. [6] trained robust models using double backpropagation to study the link between robustness and alignment in non-linear models but did not propose a new defense in their paper.
>
> All in all, our work aims to open up Jacobian’s saliency as a new avenue to boost adversarial robustness.
>
> [1] Larsen et al., 2015 “Autoencoding beyond pixels using a learned similarity metric.” arXiv:1512.09300, 2015
>
> [2] Drucker and Le Cun "Double backpropagation increasing generalization performance" IEEE Transactions on Neural Networks, 3(6):991–997, 1992.
>
> [3] Ross et al. “Improving the adversarial robustness and interpretability of deep neural networks by regularizing their input gradients.” AAAI, 2018.
>
> [4] Jakubovitz et al. “Improving dnn robustness to adversarial attacks using jacobian regularization” ECCV, pp. 514–529, 2018.
>
> [5] Simon-Gabriel et al., "First-order Adversarial Vulnerability of Neural Networks and Input Dimension" ICML, pp. 5809–5817, 2019
>
> [6] Etmann et al. “On the connection between adversarial robustness and saliency map interpretability.” arXiv:1905.04172, 2019.

---

### Public Comment · ~Zhengyu_Zhao1 · 2019-11-08
**related reference**

Hi, I just came across one arXiv article that also addresses the same question that Jacobian Regularization leads to robustness to adversarial perturbations.

It would be good to compare the differences.

https://arxiv.org/pdf/1908.02729.pdf

---

> ### Author Response · Authors · 2019-11-15
> **Response to Comment**
>
> We thank the commenter for mentioning this paper. The main contribution of [1] lies in an algorithm to efficiently approximate the input-class probability output Jacobians to minimize their Frobenius norms to improve prediction stability and adversarial robustness [1]. Similar to works like [2-4], [1]’s objective is to reduce the effect that perturbations at individual pixels have on the classifier’s prediction through the Jacobians’ norm. Different from our paper, the work in [1-4] did not aim to optimize Jacobians to explicitly resemble their corresponding images. In contrast, we propose JARN with the aim to make the Jacobian resemble input images more closely through an adversarial loss term, to explore whether this leads to improved robustness. We will add this paper to related work to make a better distinction (Section 2: Non-Adversarial Training Regularization). Furthermore, we have also included additional experiments to compare with double backpropagation [2-4] and found that JARN outperforms it in PGD attacks on CIFAR-10 (Section 4.3 and Table 3).
>
> [1] Judy Hoffman, Daniel A Roberts, and Sho Yaida. “Robust learning with jacobian regularization.” arXiv:1908.02729, 2019.
>
> [2] Drucker and Le Cun "Double backpropagation increasing generalization performance" IEEE Transactions on Neural Networks, 3(6):991–997, 1992.
>
> [3] Simon-Gabriel et al., "First-order Adversarial Vulnerability of Neural Networks and Input Dimension" ICML, pp. 5809–5817, 2019
>
> [4] Ross et al. “Improving the adversarial robustness and interpretability of deep neural networks by regularizing their input gradients.” AAAI, 2018.

---

### Author Response · Authors · 2019-11-15
**Summary of Revision**

We would like to thank all the reviewers for their insightful and constructive comments to improve the paper. We have uploaded a revision of the paper with the following updates:

> Added more discussion on why Jacobian saliency confers robustness Section 4.3.2, as suggested by Reviewer #1.

> Included results on JARN’s computational efficiency compared to adversarial training in Section 4.3.4 and Table 4, as advised by Reviewer #1, 2 & 3.

> Added experiments to show JARN’s stability across key hyperparameter changes in Section 4.3.5 and Appendix Figure 5, as recommended by Reviewer # 2 & 3.

> Added experiments to more thoroughly evaluate JARN and show its effectiveness against black-box and transfer attacks in Section 4.3.6 and Table 5, as suggested by Reviewer # 2.

> Included comparison with double backpropagation Jacobian norm regularization baseline and showed (Section 4.3 and Table 3) JARN’s approach for Jacobian saliency outperforms double backpropagation’s approach through minimizing Jacobian’s Frobenius norm values, as suggested by Reviewer # 3.

> Added further discussion on differences between JARN and closely related prior art in Section 2: “Non-Adversarial Training Regularization”, as recommended by Reviewer # 3.

---

### Public Comment · ~Haonan_Qiu1 · 2020-01-08
**Confused About Layers of Wide-Resnet.**

It's a nice paper. I just want to ask about a small problem. In experimental part 4.3, which Wide-Resnet do you use? 28-10 (Madry's setting) or 34-10? I never see the 32-10 before because WRN requires (N-4) % 6 == 0. Thanks.

---

> ### Author Response · Authors · 2020-01-08
> **Model Used Same as Madry et al.'s Setting**
>
> Thank you for your interest in our paper. We use the Wide-Resnet from Madry et al., which would be the 28-10 you mentioned. We initially called it 32-10 to be consistent with how a previous paper “Adversarial Training for Free!”, by Shafahi et al., named it. We will remove the name and emphasize our model’s similarity to Madry et al. for more clarity.

---

### Decision · Program_Chairs · 2019-12-19

**Decision:**

Accept (Poster)

**Comment:**

This paper extends previous observations (Tsipars, Etmann etc) in relations between Jacobian and robustness and directly train a model that improves robustness using Jacobians that look like images. The questions regarding computation time (suggested by two reviewers, including one of the most negative reviewers) are appropriately addressed by the authors (added experiments). Reviewers agree that the idea is novel, and some conjectured why the paper’s idea is a very sensible one. We think this paper would be an interest for ICLR readers. Please address any remaining comments from the reviewers before the final copy.